# Identification of Inhibitors of Tubulin Polymerization Using a CRISPR-Edited Cell Line with Endogenous Fluorescent Tagging of β-Tubulin and Histone H1

**DOI:** 10.3390/biom13020249

**Published:** 2023-01-29

**Authors:** Harutyun Khachatryan, Bartlomiej Olszowy, Carlos A. Barrero, John Gordon, Oscar Perez-Leal

**Affiliations:** Department of Pharmaceutical Sciences, Moulder Center for Drug Discovery, School of Pharmacy, Temple University, Philadelphia, PA 19140, USA

**Keywords:** CRISPR, tubulin, polymerization, inhibitors, high-content, imaging, screening, high-throughput, homologous, recombination, gene, tagging

## Abstract

Tubulin is a protein that plays a critical role in maintaining cellular structure and facilitating cell division. Inhibiting tubulin polymerization has been shown to be an effective strategy for inhibiting the proliferation of cancer cells. In the past, identifying compounds that could inhibit tubulin polymerization has required the use of in vitro assays utilizing purified tubulin or immunofluorescence of fixed cells. This study presents a novel approach for identifying tubulin polymerization inhibitors using a CRISPR-edited cell line that expresses fluorescently tagged β-tubulin and a nuclear protein, enabling the visualization of tubulin polymerization dynamics via high-content imaging analysis (HCI). The cells were treated with known tubulin polymerization inhibitors, colchicine, and vincristine, and the resulting phenotypic changes indicative of tubulin polymerization inhibition were confirmed using HCI. Furthermore, a library of 429 kinase inhibitors was screened, resulting in the identification of three compounds (ON-01910, HMN-214, and KX2-391) that inhibit tubulin polymerization. Live cell tracking analysis confirmed that compound treatment leads to rapid tubulin depolymerization. These findings suggest that CRISPR-edited cells with fluorescently tagged endogenous β-tubulin can be utilized to screen large compound libraries containing diverse chemical families for the identification of novel tubulin polymerization inhibitors.

## 1. Introduction

Cancer constitutes the second most common cause of death following cardiovascular diseases worldwide [1]. Despite significant progress in the understanding, diagnosis, prevention, and treatment of cancer, it remains a major threat to global human health. As the world’s population continues to expand and age, the World Health Organization estimates that 13.2 million cancer-related deaths will occur worldwide by 2030, compared to 7.6 million in 2008 [2].

Of the many approaches used to treat cancer, chemotherapy is one of the most common and effective tools used by oncologists [3]. In chemotherapy, drugs that disrupt microtubule/tubulin dynamics are commonly employed [4]. These drugs directly interfere with the cell’s microtubule system as opposed to acting on the DNA [5]. Microtubules are cytoskeletal structures that allow a cell to maintain its shape. They are also critical in mitotic chromosome separation and create the mitotic spindle, and are thus essential for the successful completion of cell division [6].

Microtubules composed of alpha and β-tubulin heterodimers that wind together to form a hollow cylinder of 24 nm in diameter [7]. These structures can rapidly grow via polymerization and shrink via depolymerization [8]. Epipodophyllotoxins, taxanes, vinca alkaloids, epitholones, and macrolides are among the most common chemotherapeutic drugs that impair microtubule activity [9]. Microtubule targeting agents work by interfering with the dynamics of the microtubule, either inhibiting polymerization or depolymerization.

Microtubule targeting agents interact with tubulin via at least four different binding sites: laulimalide, taxane/epothilone, vinca alkaloid, and colchicine [10]. Polymerization inhibitors mainly act on the vinca alkaloid and colchicine binding sites, while depolymerization inhibitors act on the taxane binding site [11]. Vinca alkaloids, unlike colchicine, attach directly to the microtubule. They do not form a complex with soluble tubulin or copolymerize to form the microtubule, but they are capable of causing a conformational shift in tubulin in the context of tubulin self-association [12].

While tubulin polymerization inhibitors have made great progress in the treatment of many forms of cancer, they have several drawbacks, including hematopoietic and neurologic toxicities, inconvenience formulation, and resistance. Therefore, identifying novel tubulin polymerization inhibitors is necessary to combat these barriers [9].

Traditionally, compounds that are inhibitors of tubulin polymerization have been found using pure tubulin in vitro [13] or procedures that require the immunostaining of cells [14]. Additionally, the study of tubulin dynamics in live cells upon drug or genetic manipulation has been conducted using cells that overexpress recombinant fluorescently labeled histone and tubulin proteins [15,16]. We recently reported a new technology that facilitates using CRISPR genome editing for developing cell lines called FAST-HDR [17]. This technology allows the generation of cell lines by tagging multiple endogenous genes with fluorescent proteins. This approach facilitates the study of endogenous proteins in live cells without using cell fixing, immunostaining or recombinant protein overexpression [17].

This work demonstrates the use of a cell line with endogenous tagging with fluorescent proteins of β-tubulin and histone H1 to identify novel tubulin polymerization inhibitors. We screened a library of kinase inhibitors and identified and characterized three compounds as tubulin polymerization inhibitors using high-content imaging (HCI) analysis of live cells.

## 2. Materials and Methods

### 2.1. Cell Line Development

HeLa cells with endogenous tagging with fluorescent proteins of β-tubulin (mClover3), histone H1 (mTagBFP2), and p62-SQSTM1 (mRuby3) were developed with CRISPR and the FAST-HDR vector system, and were recently described [17]. These cells are commercially available from ExpressCells, Inc. A clonal cell line was derived by single cell sorting with the Hana Single Cell Sorter Instrument (Namocell) using the green fluorescent channel and sorting into a 96 well plate. Single cell clones were analyzed with a Spark Cyto plate reader (Tecan) using whole-well imaging for selecting clones derived from a single cell. A clone with uniform and bright β-tubulin fluorescence was selected to expand a cell line for this study.

### 2.2. High-Content Imaging, Compound Screening and Time-Lapsed Microscopy

The CRISPR-edited HeLa cell line described above was used to screen a small compound library of 429 kinase inhibitors (SelleckChem, Houston, TX, USA) for high-content imaging drug screening. The positive control compounds (colchicine and vincristine) as well as the identified hits in this study (ON-01910, KX2-391, and HMN-214) were acquired for further validation from Cayman Chemicals in powder form and dissolved with DMSO. All confocal images in this work were acquired with live cells grown in Dulbecco’s Modified Eagle Medium (DMEM) without phenol red (Thermo Fisher Scientific, Waltham, MA, USA) supplemented with 10% Fetal Bovine Serum (100 uL). The 384-well CellCarrier Ultra plates (PerkinElmer, Waltham, MA, USA) were seeded with 5 × 10^3^ cells per well and 16 h later were treated with the library compounds at a final concentration of 1 uM using the PerkinElmer JANUS Automated Workstation. The compounds’ effects were assessed 24 h later with an Operetta CLS Confocal High Content Imaging System (PerkinElmer, Waltham, MA, USA) and the following parameters: excitation and emission filter combinations: mTagBFP2 (Ex-405, Em-440), mClover3 (Ex-480, Em-513); acquisition time: 500 ms per channel. Two images (with two different channels) were acquired with the same coordinates for each well using a 40X water immersion objective. On average, between 60 and 100 cells were analyzed per well. As negative controls 32 wells in each 384-well plate were only treated with dimethyl sulfoxide (DMSO) at 1% final concentration.

For time-lapse microscopy, cells were kept under environmentally controlled conditions (37 °C, 5% CO_2_), and images were acquired every three minutes with the same filter combinations described above. All conditions were analyzed in triplicates, and the experiments were repeated three independent times. All compounds under study were administered 30 min after the start of image acquisition in a concentration range between 1 nM and 4 uM.

### 2.3. Image Data Analysis

For image data analysis, we used Harmony Software 4.8 (Perkin Elmer). The analysis was conducted using the following steps within the Analysis module of Harmony Software: (1) identifying the cell nucleus with the “Find Nuclei” algorithm using the Histone H1-mtagBFP2 channel, (2) detecting the cytoplasm boundaries of the identified cell nucleus using the “Find Cytoplasm” algorithm with the TUBB-mClover3 channel, and (3) analyzing the texture properties of the cytoplasm of each cell using the Haralick analysis option with default settings. The Haralick texture analysis option provides values for pixel correlation, contrast, homogeneity, and sum variance. For this study, we only evaluated changes in the mean pixel homogeneity values per each well.

### 2.4. Molecular Docking

We used the x-ray diffraction crystallography protein structure of β-tubulin from *bos taurus* (protein data bank code: 4O2B) for molecular docking experiments. This *bos taurus* tubulin isoform has a 100% homology to human β-tubulin (TUBB). The blind molecular docking was performed with CB-Dock version 2, a cavity detection-guided protein–ligand blind docking web server [18] that uses Autodock Vina (version 1.1.2, Scripps Research Institute, La Jolla, USA). The SDF structure files of the tested compounds (colchicine, vinblastine, paclitaxel, ON-01910, KX2-391, and HMN-214) were downloaded from PubChem. The molecular blind docking was performed by uploading the 3D structure PDB file of β-tubulin into the server with the SDF file of each compound. For analysis, we selected the docking poses with the strongest Vina score. The generated PDB files of the molecular docking of each compound were visualized with the Maestro software (Schrödinger, Inc., New York, NY, USA) and were compared against the experimentally validated X-ray structures of the interaction of three drugs with β-tubulin (Colchicine: 4O2B, Paclitaxel:1JFF, Vinblastine:5J2T). The Redocking analysis of the compounds (ON-01910, KX2-391, and HMN-214) with the colchicine binding site of β-tubulin was performed with OEDocking from OpenEye Scientific. For this, the X-ray structure of β-tubulin interacting with colchicine (4O2B) was used to build a box grid with an automated molecular cavity detection routine for targeted docking around the colchicine binding site. Docking was done with the SDF files of ON-01910, KX2-391, and HMN-214 obtained from PubChem. The ligand-protein interactions were evaluated in Chimera X [19] and Maestro software.

### 2.5. Reproducibility and Statistical Analysis

All figures in this work are representative of at least three independent experiments. The comparison of multiple groups was performed using one-way analysis of variance with Dunnett’s test within GraphPad Prism (version 9) (GraphPad Software, La Jolla, CA, USA); *p* values < 0.05 were considered statistically significant. Differences between control and treated samples in time-dependent experiments were found using two-way analysis of variance within GraphPad Prism.

## 3. Results

### 3.1. Detecting the Inhibition of Tubulin Polymerization in Live Cells without Using Antibodies or Chemical Staining

The analysis of phenotypic changes caused by compounds that alter tubulin polymerization usually requires the use of antibodies, chemical staining, or promoting the overexpression of fluorescent tubulin [20]. However, some of these techniques prevent the analysis of live cells to evaluate changes that happen over time or can induce microscopical artifacts by uncontrolled overexpression. As an alternative approach, here, we conducted the exploration using a recently developed HeLa cell line with endogenous tagging of native proteins with fluorescent proteins via CRISPR-Cas9 [17]. In this cell line, endogenous histone H1 is tagged with a blue fluorescent protein (mTagBFP2) and allows the detection of the nucleus. Additionally, β-tubulin (TUBB) is tagged with a green fluorescent protein (mClover3) and facilitates the identification of the microtubules in the cytoplasm. In order to evaluate the potential of this cell line for detecting phenotypic changes in tubulin polymerization by HCI analysis, an analysis routine with the Harmony software (version 4.8) was developed. The routine was set up to detect individual cells in each acquired field by analyzing two independent fluorescent channels. In the blue channel, the software detects the nuclei, where a nuclear protein is labeled with a blue fluorescent protein, using the Find Nuclei algorithm. In the green channel, the software defines the cytoplasm boundaries of every nucleus with the Find Cytoplasm algorithm by detecting tubulin that is labeled with a green fluorescent protein. Once the software identifies each cell, it can analyze texture properties of the cytoplasm of every cell (Figure 1A).

It was then identified how these cells respond to tubulin polymerization inhibitors by first treating the cells with two different tubulin polymerization inhibitors: vincristine and colchicine. Phenotypic changes were identified, where the normal pattern of the microtubules disappears in treated cells (Figure 1B). Next, it was evaluated if these phenotypic changes translate to changes in the texture of the image of tubulin for each cell for automatic detection during compound screening. For that, the Haralick texture homogeneity algorithm was used (In Harmony Software) because it has been shown to be an optimal tool for analyzing the texture of microtubules [21]. This algorithm analyzes the homogeneity of the pixels in an image to determine regularity [22] and can be implemented in multiple commercial and open-source software packages, including MatLab [23] and CellProfiler [24]. The analysis of the images from cells treated with known tubulin polymerization inhibitors or controls shows that the Haralick texture homogeneity values increase for vincristine and colchicine, and these changes were statistically significant (Figure 1B).

### 3.2. Identifying Tubulin Polymerization Inhibitors in a Library of Kinase Inhibitors

The HeLa cell line with endogenous tagging of histone H1 and β-tubulin was used for screening a library of 429 kinase inhibitors to identify tubulin polymerization inhibitors using HCI analysis. The cells were plated onto 384 well plates, treated with the library of kinase inhibitors for 16 h, and directly analyzed with an automated high-content imager (Figure 2A). The obtained images were analyzed with the procedure depicted in Figure 1A. The Haralick texture homogeneity values of cells treated with all the compounds were determined to identify potential inhibitors of tubulin polymerization. One point of interest was compounds that could induce changes in the Haralick texture homogeneity values that were higher than the mean plus three times the standard deviation of all the compounds. According to the data (Figure 2B), four compounds were identified as positive hits; however, after manual inspection of the images of these four wells, only three compounds (ON-01910, HMN-214, and KX2-391) were found to be positive hits.

### 3.3. Validating Kinase Inhibitors as Tubulin Polymerization Inhibitors

To confirm that ON-01910, HMN-214, and KX2-391 are inhibitors of tubulin polymerization, a dose response analysis was performed for detecting the IC_50_ dose required to find phenotypic changes in tubulin polymerization. First, the optimal amount of time required to detect tubulin polymerization inhibition of modified HeLa cells treated with colchicine was determined by conducting time-lapse imaging for 16 h. This showed that the tubulin polymerization inhibition process is fast and can be detected in the first three hours of treatment. The Haralick homogeneity values of cells treated for three hours were obtained with the three hits compounds and colchicine in a concentration range between 1 nM and 4 μM. The validation showed that the three compounds can inhibit tubulin polymerization (Figure 3A). It was found that the three compounds have different IC_50_ potencies, with KX2-391 having the highest potency, followed by ON-01910 and then HMN-214 (Figure 3B). Additionally, it was observed that colchicine can inhibit tubulin polymerization at lower doses (IC_50_ = 58nM) compared to the three compounds. Notably, a recent report has described that colchicine has an IC_50_ value of 10.65nM for inhibiting the polymerization of purified microtubules in vitro [25]. Our results suggest that analyzing changes in the Haralick homogeneity values of microtubules in live cells by high-content imaging can be used to determine the potency of compounds that alter tubulin polymerization.

Next, it was evaluated how rapidly ON-01910, KX2-391, and colchicine affected tubulin polymerization by tracking treated cells over a period of three hours by acquiring images in the two fluorescent channels every three minutes. After analyzing the time-lapsed images with the routine described in Figure 1A, the dynamic Haralick homogeneity values (Figure 4) were obtained. It was confirmed that the three compounds inhibit tubulin polymerization rapidly, and the changes can be detected in the first 20 min of treatment. The results indicate that treatment with colchicine yields the highest change in Haralick texture homogeneity, followed by KX2-391 and then ON-01910. In the control group, the cells maintain the structure of the tubulin and continue to divide; however, for all the treatment groups, the tubulin structures rapidly disappear (Appendix A).

### 3.4. Molecular Docking of Kinase Inhibitors That Inhibit Tubulin Polymerization

Another aim was to evaluate the potential interaction of the hit compounds (ON-01910, KX2-391, and HMN-214) with β-tubulin and to confirm that the effect of tubulin polymerization inhibition is achieved by directly blocking tubulin function. We evaluated the potential interaction of these compounds with β-tubulin using molecular docking, which is used to predict the predominant binding mode of a ligand with a protein of known three-dimensional structure [26].

To test the approach, blind molecular docking with known binders of β-tubulin (colchicine, vinblastine, and paclitaxel) was performed using the three-dimensional structure of β-tubulin from *bos taurus*, which has 100% sequence identity with human β-tubulin (TUBB). The predicted interactions match the location of experimental x-ray crystallography structures (Figure 5A). Following this validation, a blind docking prediction was performed of the interaction of the three kinase inhibitors: ON-01910, KX2-391, and HMN-214 with β-tubulin. The analysis suggests that these compounds all interact with the colchicine binding site (Figure 5B). The molecular docking software ranks the binding modes according to the Vina score, an empirical scoring function that assesses the contributions of a variety of different factors and determines the affinity of the protein–ligand interaction [27]. Based on the rankings, the smaller the value (larger negative number) of the Vina score, the more accurate the predicting interaction. Based on the docking results for colchicine, ON-01910, HMN-214, and KX2-391, the Vina scores were −7.0, −7.0, −8.6, and −8.0, respectively. To validate the specific interactions of the new compounds with the colchicine binding site, we performed redocking analysis with a focused grid around the pocket of colchicine using OEDocking software (Figure 6A). We found that there were seven amino acids in β-tubulin (LEU 242, CYS 241, LYS 352, ASN 258, LEU 255, LEU 248, ALA 250) that commonly participated in the interaction with colchicine and all the new compounds (Figure 6B). A recent review has highlighted that the amino acids CYS 241, ALA 250, ASP 251, and LEU 252 play a crucial role in the tight binding of colchicine with β-tubulin [28]. Notably, many of these amino acids also participate in interactions with the new compounds, as depicted in Figure 6B. The docking analysis reveals several key differences between colchicine and the new compounds. For instance, KX2-391 is a compound with an elongated shape that can protrude out of the binding pocket, as illustrated in Figure 6A. Despite this, it is more effective in inhibiting tubulin polymerization than the other two compounds. ON-01910 has an aromatic ring with three methoxy-substituted groups that is similar to that of colchicine, with the exception that the methoxy groups are located in different positions, as shown in Figure 6B. This aromatic ring in ON-01910 interacts with most of the amino acids that are key for the binding of colchicine. However, its more planar structure may affect the strength of its interaction within the colchicine pocket. Lastly, HMN-214 contains two polar groups within the non-polar binding cavity of tubulin, which may make the binding less stable and can explain why it is the least potent in inhibiting tubulin polymerization despite its more compact structure compared to the other two compounds.

## 4. Discussion

Microtubules play a crucial role in several biological processes, including cellular shape maintenance, cell motility, and mitosis [29]. Due to the clinical efficacy of some microtubule-targeting medicines for treating cancer, microtubules continue to be promising targets for developing novel anti-cancer drugs [30].

This work describes a simplified process to identify tubulin polymerization inhibitors with live cells. The method relies on the use of a CRISPR-modified cell line that contains fluorescent proteins tagged to endogenous β-tubulin and histone H1. This cell line allows the direct use of high-content imaging analysis for cellular segmentation and microtubule network detection without requiring cell fixing or immunostaining. With this method, the three compounds (ON-01910, KX2-391, and HMN-214) were identified in a group of 429 kinase inhibitors as tubulin polymerization inhibitors. Previous reports that used alternate methodologies validate these findings for two of these compounds. ON-01910 was reported to show tubulin destabilization activity after comparing changes in cells with genome-wide upregulation/downregulation via a CRISPR activation/inhibition approach and treated with this compound [31]. In addition, KX2-391 was initially developed as a low-nanomolar inhibitor of Src tyrosine kinase for cancer treatment; however, further studies confirmed that at higher concentrations, KX2-391 has tubulin polymerization inhibition activity [32]. KX2-391 went to clinical trials as a topical tubulin polymerization inhibitor for the treatment of actinic keratosis and received FDA approval in December 2020 [33].

Two of the three compounds identified in this study (ON-01910 and KX2-391) are potent (<10 nM IC_50_) kinase inhibitors of their respective target, while HMN-214 is an inactive prodrug that requires liver metabolic transformation before becoming a kinase inhibitor [34]. The capability of these compounds to inhibit tubulin polymerization seems to be independent of their function for inhibiting their kinase target. This assumption is suggested by showing that their capability to inhibit tubulin polymerization only occurs at high concentrations (>250 nM). In addition, HMN-214 does not have kinase inhibitory activity before going through liver transformation but can inhibit tubulin polymerization at high concentrations.

The blind molecular docking analysis indicates that these three compounds can interact directly with the colchicine binding site of β-tubulin, thus suggesting a similar mechanism of action for inhibiting tubulin polymerization. The blind molecular docking was done using CB-Dock version 2, a cavity detection-guided protein–ligand blind docking web server [18]. The current version of CB-Dock 2 has a calculated 85.9% prediction accuracy [18], thus indicating that there is a high probability that the right binding pocket was identified.

This study validates the simple use of a CRISPR-edited cell line for identifying tubulin polymerization inhibitors via high-content imaging analysis. This method is considerably cheaper and less laborious than previous methods due to the elimination of the required steps for cellular fixing and immunostaining. Additionally, this method allows live cell analysis of tubulin polymerization for evaluating kinetic changes. We foresee the use of cell lines with endogenous tagging of tubulin with fluorescence proteins for screening larger and diverse libraries for discovering novel tubulin polymerization inhibitors. We also anticipate that this type of cell lines could be good models to validate novel tubulin polymerization inhibitors discovered by virtual compound screening.

## Figures and Tables

**Figure 1 biomolecules-13-00249-f001:**
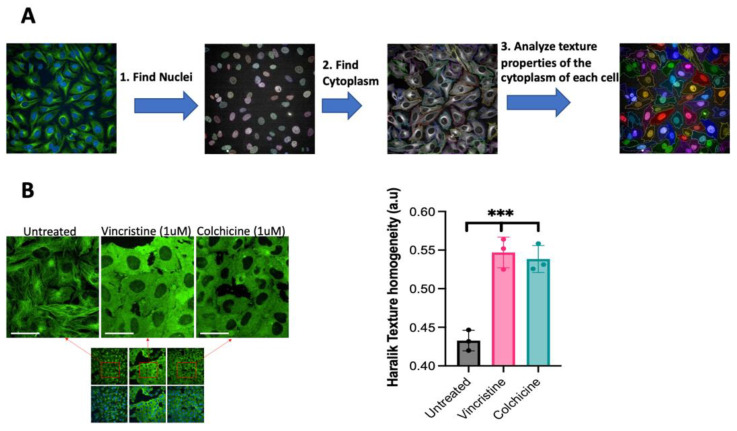
Detection of tubulin polymerization changes in live cells with HCI analysis. (**A**) The changes in tubulin polymerization were detected and quantified in each microscopic image with the following image analysis workflow: first, detection of the nuclei of every cell in the image using the blue fluorescent channel for detecting histone H1-mTagBFP2. Second, detection of the cytoplasm of every nuclei that was identified in the first step using the green fluorescent channel for detecting β-tubulin-mClover3. Third, quantitate the image texture properties of the cytoplasm of every cell. (**B**) Confocal microscopy images of live CRISPR-edited HeLa cells with fluorescent tagging of histone H1 and β-tubulin. The tubulin polymerization inhibition was detected after treatment with known tubulin polymerization inhibitors (vincristine and colchicine). The changes in tubulin polymerization with each treatment were quantified with the Haralick texture homogeneity algorithm and are shown on the right panel. Differences were analyzed by one-way analysis of variance (ANOVA), and *p*  < 0.05 was considered statistically significant. *** *p* < 0.001; error bars represent ± standard deviation (*SD*) of images from triplicate wells. These data are representative of three independent experiments; scale bar = 15 μm.

**Figure 2 biomolecules-13-00249-f002:**
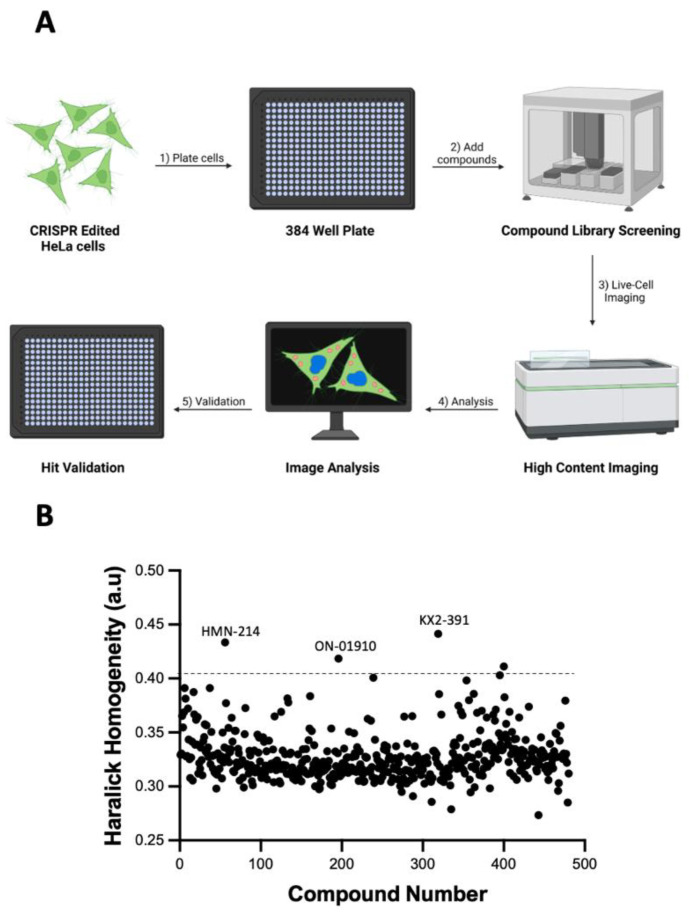
Kinase inhibitor library screening with HCI analysis of live cells. (**A**) Diagram of the steps required to detect phenotypic changes of CRISPR-edited HeLa cells after treatment with a library of kinase inhibitors. The cells are plated into 384 glass-bottom plates and treated with the compounds 16 h later. Following a 24 h incubation period, the cells are imaged directly without cell fixing or immunostaining. The images are analyzed to detect changes in the Haralick homogeneity values as described in Figure 1A. Lastly, the positive hits are further validated. (**B**) Changes in the Haralick texture homogeneity values of cells treated with a library of 429 kinase inhibitors. Compounds that change the Haralick texture homogeneity higher than the mean of all the compounds plus three standard deviations (dashed line) were considered positive hits for further evaluation. Three compounds (ON-01910, KX2-391, and HMN-214) were manually checked and selected for validation. These data are representative of three independent experiments.

**Figure 3 biomolecules-13-00249-f003:**
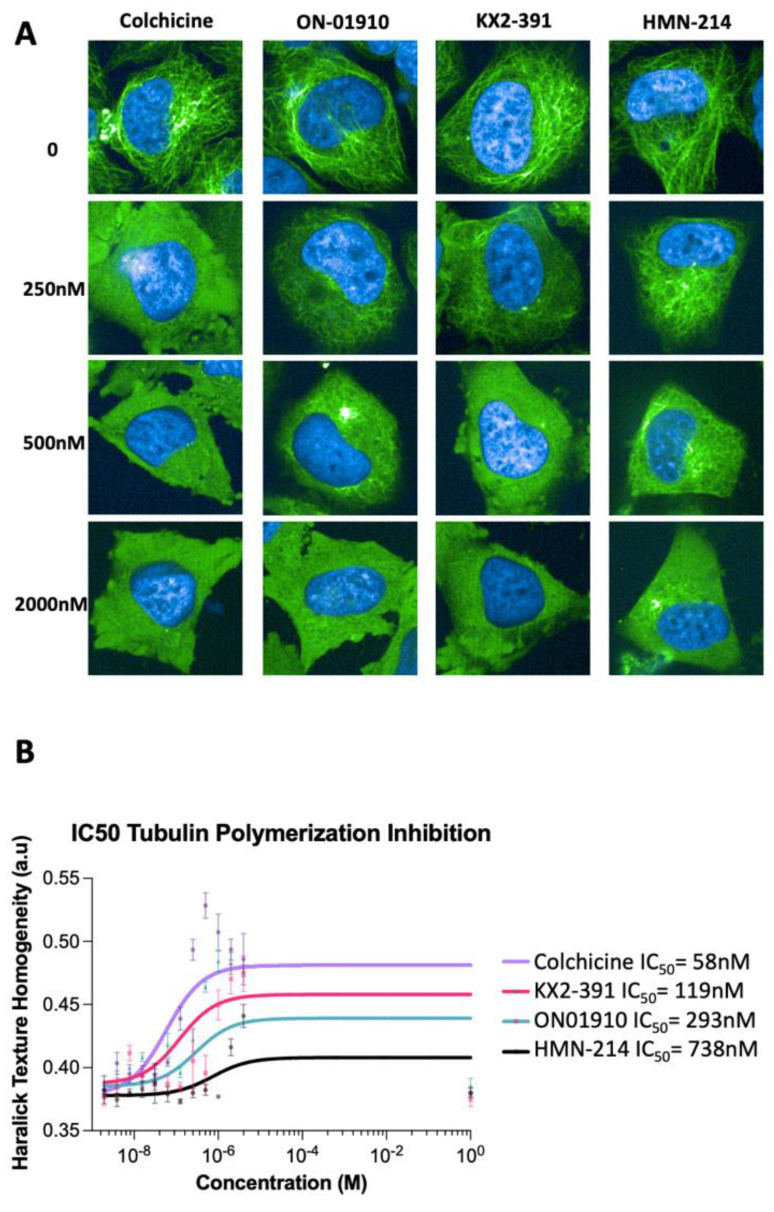
Validation of ON-01910, KX2-391, and HMN-214 as tubulin polymerization inhibitors. (**A**) Representative images of cells treated with three kinase inhibitors identified as potential tubulin polymerization inhibitors. The cells were treated in triplicates in a range of concentrations between 1 nM and 4000 nM. (**B**) The Haralick texture homogeneity values of cells treated with the three compounds (1 nM to 4000 nM) was used to determine the phenotypic half maximal tubulin polymerization inhibitory concentration (IC_50_). Each data point is n = 3; these data are representative of three independent experiments.

**Figure 4 biomolecules-13-00249-f004:**
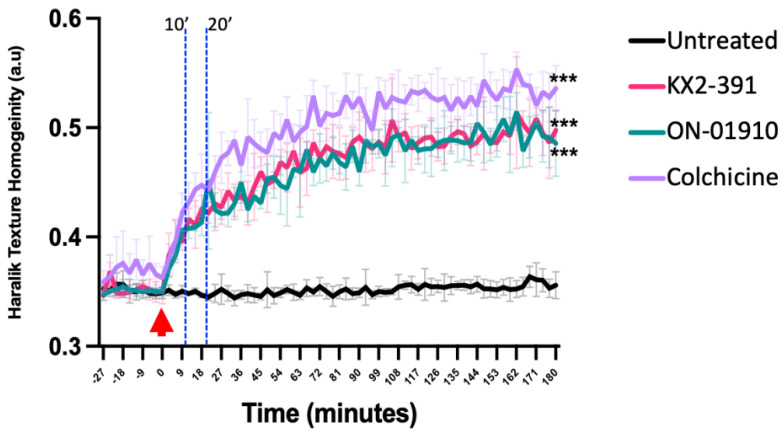
Live cell tracking of the inhibition of tubulin polymerization by detecting changes in the Haralick homogeneity values. HeLa cells with endogenous tagging of β-tubulin and histone H1 were treated with ON-01910, KX2-391, and colchicine as a positive control to detect the tubulin polymerization inhibition over a period of three hours. Imaging capture (every three minutes) started 30 min before adding the compounds and continued for a total of three hours. The red arrow indicates the start of the compound treatment. The two vertical dashed lines indicate the 10 and 20 min timepoints. Differences between drug treatments vs. untreated control were analyzed by two-way ANOVA, and *p* < 0.05 was considered statistically significant. *** *p* < 0.001; error bars represent ± *SD* of triplicate wells. This figure is representative of three independent experiments.

**Figure 5 biomolecules-13-00249-f005:**
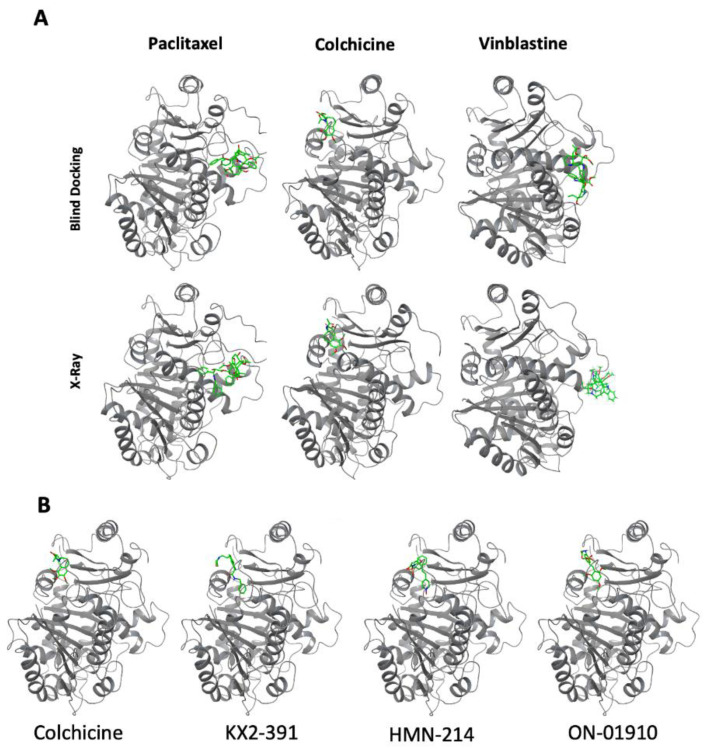
Identification of the molecular binding site in β-tubulin of kinase inhibitors that inhibit tubulin polymerization using blind docking. (**A**) A comparison of molecular blind docking vs. experimental X-Ray structure data of the 3D binding site in β-tubulin of drugs that inhibit tubulin polymerization (colchicine, vinblastine) or a drug that can stabilize microtubules (paclitaxel). Blind docking predictions were made with CB-Dock 2 web server. (**B**) A comparison of the β-tubulin predicted binding site of kinase inhibitors (KX2-391, HMN-214, ON-01910) that can inhibit tubulin polymerization against the binding site of colchicine.

**Figure 6 biomolecules-13-00249-f006:**
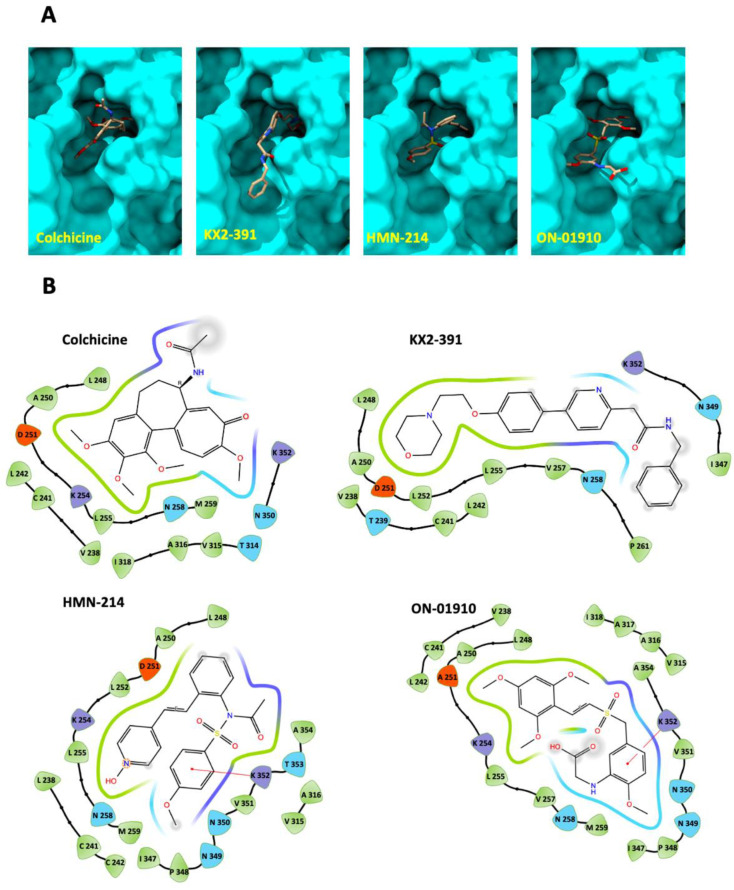
Docket pose against the β-tubulin colchicine binding site of kinase inhibitors that inhibit tubulin polymerization. (**A**) Docket pose of KX2-391, HMN-214, and ON-01910 against the colchicine binding site in β-tubulin and generated with AEDocking. (**B**) Maestro software was used to visualize the amino acids surrounding the β-tubulin binding site of colchicine, KX2-391, HMN-214, and ON-01910.

## Data Availability

Not applicable.

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
