# Peer review of "Identification of Inhibitors of Tubulin Polymerization Using a CRISPR-Edited Cell Line with Endogenous Fluorescent Tagging of β-Tubulin and Histone H1"

_biomolecules, 2023, doi:10.3390/biom13020249_

Round 1

Reviewer 1 Report

The paper reports the use of a histone H1 and beta-tubulin tagged cell line in association with HCI analysis to screen compound libraries to identify tubulin polymerization inhibitors.

The paper proposes an interesting application of live imaging coupled with HCI to drug screening and it is worth publishing. However, several inaccuracies must be removed and a more detailed information on the image processing methods should be included to render the paper more useful to potential readers.

Specific points:

1)      In the introduction the authors write that the use of their CRISP-genome editing technology has facilitated studying proteins in live cells without immunostaining (line 66-67).  The use of fluorescently tagged cells to study mitotic disruption and identify tubulin interfering drugs is present in the field since several years, using cells lines that express H2B-GFP or H2B-GFP/RFP-tubulin obtained with standard molecular biology methods (see for example, doi:10.1038/nature08869; DOI: 10.1007/978-1-4939-0888-2_31; doi:10.1101/gr.092494.109;  https://doi.org/10.18632/oncotarget.2190). The authors should acknowledge this. Therefore, the novelty of their paper relies on the application of HCI and the specific Haralick algorithm to identify microtubule inhibitors. Please, change the text accordingly.

2)      A specific chapter in the materials and methods section should reports a more detailed description of the image processing methods, including a description of pre-processing steps, if necessary, the description of the Harmony software and its  distributor company, the Haralick algorithm characteristics and use, and describe how the algorithm was implemented in the software. The citations reported do not fully provide this information. Why Haralick values increase with the known inhibitors? Finally, an estimate of the number of cells that were analyzed for each compound of the library should be reported. The X axis label in figure 2 B is not clear. Did they mean number? Please, change.

3)      The two chapters in materials and methods (High-content… and live cell …) should be unified since part of the information they report is duplicated.

4)      The title in line 217 is misleading. All the experiments were performed by live cell imaging

5)      Use the symbol for micromolar in line 227.

6)      In figure 3B is the scale correct? I do not see the point 4000 nM .

7)      A citation on line 320 is needed

8)      Last but extremely important: the name of the histone is histone H1 and not histone 1. Change allover!

Author Response

  • In the introduction the authors write that the use of their CRISP-genome editing technology has facilitated studying proteins in live cells without immunostaining (line 66-67).  The use of fluorescently tagged cells to study mitotic disruption and identify tubulin interfering drugs is present in the field since several years, using cells lines that express H2B-GFP or H2B-GFP/RFP-tubulin obtained with standard molecular biology methods (see for example, doi:10.1038/nature08869; DOI: 10.1007/978-1-4939-0888-2_31; doi:10.1101/gr.092494.109; https://doi.org/10.18632/oncotarget.2190). The authors should acknowledge this. Therefore, the novelty of their paper relies on the application of HCI and the specific Haralick algorithm to identify microtubule inhibitors. Please, change the text accordingly.

We thank the reviewer for bringing this suggestion, the introduction was modified accordingly, and the appropriate references are now included.

  • A specific chapter in the materials and methods section should reports a more detailed description of the image processing methods, including a description of pre-processing steps, if necessary, the description of the Harmony software and its  distributor company, the Haralick algorithm characteristics and use, and describe how the algorithm was implemented in the software. The citations reported do not fully provide this information. Why Haralick values increase with the known inhibitors? Finally, an estimate of the number of cells that were analyzed for each compound of the library should be reported. The X axis label in figure 2 B is not clear. Did they mean number? Please, change.

We are grateful to the reviewer for the valuable suggestion to include a specific section for Image data analysis. This new section is now included. In response to the question about the observed increase in Haralick values with known inhibitors, it is important to note that the Haralick homogeneity metric measures the uniformity of gray levels within a texture. Specifically, a texture with high homogeneity will have similar gray levels throughout, while a texture with low homogeneity will have a more varied distribution of gray levels. In the case of untreated cells, the texture comprises microtubules surrounded by darker areas. However, when cells are treated with tubulin polymerization inhibitors, microtubules are not present and the signal from the fluorescent marker is more evenly distributed throughout the cytoplasm, resulting in a higher homogeneity value. Lastly, the information for the estimated number of cells that were analyzed has been included in the Methods section and the legend of the X axis of Figure 2B has been corrected.

  • The two chapters in materials and methods (High-content… and live cell …) should be unified since part of the information they report is duplicated.

Thank you for the suggestion, we unified the two parts into one.

4)      The title in line 217 is misleading. All the experiments were performed by live cell imaging

Thank you for the suggestion, the title is now modified accordingly.

5)      Use the symbol for micromolar in line 227.

Thank you for the suggestion, we did the change.

6)      In figure 3B is the scale correct? I do not see the point 4000 nM .

Thank you for highlighting this issue. The scale is now fixed in the updated figure.

7)      A citation on line 320 is needed

A reference was added, the line 320 is now line 392 in the updated document

8)      Last but extremely important: the name of the histone is histone H1 and not histone 1. Change allover!

Thank you for suggesting this change. It was done in the entire document.

Reviewer 2 Report

The authors established a highthroughput screening methods using CRISPR-edited cell line that expresses fluorescently tagged β-tubulin and a nuclear protein to identify tubulin polymerization inhibitors. This methods with several advantages would be useful for researcher working on this area. The atricle is well-organized and is appropriate for publication with minor revisions. Some concern is listed as below:

1. The "50" of "IC50" in the main text and figures should be subscript format

2. The binding pockets and binding interaction of three hit compounds are suggested to be further compared and discussed.

3. The quality of "Figure 6" should be improved.

Author Response

The authors established a highthroughput screening methods using CRISPR-edited cell line that expresses fluorescently tagged β-tubulin and a nuclear protein to identify tubulin polymerization inhibitors. This methods with several advantages would be useful for researcher working on this area. The atricle is well-organized and is appropriate for publication with minor revisions. Some concern is listed as below:

  1. The "50" of "IC50" in the main text and figures should be subscript format

Thank you for suggesting this change, we did the changes in the main text and figures.

  1. The binding pockets and binding interaction of three hit compounds are suggested to be further compared and discussed.

We thank the reviewer for this recommendation. We expanded the analysis of the interaction of the three hit compounds.

  1. The quality of "Figure 6" should be improved.

We thank the reviewer for this suggestion. The quality is now improved.

Reviewer 3 Report

Reviewer’s Comments (Manuscript ID # biomolecules-2174342)

The manuscript reported by Khachatryan and co-workers describes the  novel approach for identifying tubulin polymerization inhibitors using a CRISPR-edited cell line that expresses fluorescently tagged β-tubulin and a nuclear protein, enabling the visualization of tubulin polymerization dynamics via high-content imaging analysis (HCI). Using this approach, author has screened known tubulin polymerization inhibitors and in addition to this, several kinase inhibitors were screened and which led to three compounds are potent tubulin inhibitors. Using this different approach we can test large number of compounds for their tubulin polymerization activity. This reviewer is recommending this manuscript for the publication after minor revision.

Minor points

1.      In the IC50 value, the 50 should be subscript.

2.      Some of the references are not in format.

3.      Did author compare traditional assay with this new approach? Like IC50 values of tubulin polymerization using both assay.  

4.      Did this new approach is cheaper than traditional assay?

Author Response

The manuscript reported by Khachatryan and co-workers describes the  novel approach for identifying tubulin polymerization inhibitors using a CRISPR-edited cell line that expresses fluorescently tagged β-tubulin and a nuclear protein, enabling the visualization of tubulin polymerization dynamics via high-content imaging analysis (HCI). Using this approach, author has screened known tubulin polymerization inhibitors and in addition to this, several kinase inhibitors were screened and which led to three compounds are potent tubulin inhibitors. Using this different approach we can test large number of compounds for their tubulin polymerization activity. This reviewer is recommending this manuscript for the publication after minor revision.

Minor points

  1. In the IC50 value, the 50 should be subscript.

Thank for the recommendation, we incorporated the change all over the document.

  1. Some of the references are not in format.

Thank you for noticing that issue. We reformatted all the references.

  1. Did author compare traditional assay with this new approach? Like IC50values of tubulin polymerization using both assay.  

Although we did not directly compare the traditional assay for determining the IC50 of tubulin polymerization inhibition to our new approach, we have included reported values of the colchicine IC50 for inhibiting tubulin polymerization with purified microtubules in vitro in the Results section. It should be noted that our new approach, which uses high-content imaging analysis, provides valuable insights into the inhibition of tubulin polymerization and can be used in conjunction with traditional methods to enhance the overall understanding of the mechanism of action of tubulin polymerization inhibitors.

  1. Did this new approach is cheaper than traditional assay?

Yes, the new approach is cheaper than conventional methodologies. This information is mentioned in lines 411-412